# Whole Genome Sequencing for Studying *Helicobacter pylori* Antimicrobial Resistance

**DOI:** 10.3390/antibiotics12071135

**Published:** 2023-06-30

**Authors:** Belén Fernández-Caso, Ana Miqueleiz, Teresa Alarcón

**Affiliations:** 1Department of Microbiology, Hospital Valle del Nalón, 33920 Langreo, Spain; belen.fernandezcca@sespa.es; 2Department of Microbiology, Hospital Universitario de Navarra, 31008 Pamplona, Spain; ana.miqueleiz.zapatero@navarra.es; 3Department of Microbiology, Hospital Universitario La Princesa, 28006 Madrid, Spain

**Keywords:** *Helicobacter pylori*, antibiotic, resistance, whole genome sequencing

## Abstract

Antibiotic resistance (AMR) is an alarming concern worldwide and *Helicobacter pylori*, one of the most prevalent bacteria, is not an exception. With antibiotics being its primary therapy, increasing resistance leads to a higher rate of treatment failure. Understanding the genomic mechanisms of resistance to clarithromycin, levofloxacin, metronidazole, amoxicillin, tetracycline, and rifampicin through next-generation sequencing-based molecular tools, such as whole genome sequencing (WGS), can be of great value, not only to direct a patient’s treatment, but also to establish and optimize treatment guidelines according to the local epidemiology and to avoid the use of inappropriate antibiotics. WGS approaches allow us to gain insight into the genomic determinants involved in AMR. To this end, different pipelines and platforms are continuously being developed. In this study, we take a more detailed view of the use and progression of WGS for in-depth study of *H. pylori*’s AMR.

## 1. Introduction

Nowadays, antimicrobial resistance (AMR) is one of the biggest threats to health worldwide. In this scenario, *Helicobacter pylori* (*H. pylori*) has emerged as an alarming bacterium regarding its AMR rates [1].

*H. pylori*, one of the most prevalent pathogens, is well-known for affecting more than 50% of the world’s population due to its ability to survive in the acidic environment of the stomach. The persistence of these bacteria in the gastric microbiota can lead to gastrointestinal diseases such as peptic ulcer disease, chronic gastritis, or even gastric adenocarcinoma [2].

Treatment of *H. pylori* relies on antibiotic therapy. The most commonly used antibiotics are clarithromycin, metronidazole, levofloxacin, amoxicillin, tetracycline, and rifabutin. Multiple regimens combining proton pump inhibitors with two or three antibiotics and/or a bismuth component have been evaluated [3]. However, the efficacy of most of these antibiotics has substantially decreased [4] and *H. pylori* eradication has become hard to accomplish when using first-line therapies. Therefore, there is an undeniable need to collect accurate local antimicrobial susceptibility data and to further investigate the mechanism underlying AMR to achieve optimal treatments.

The advent of molecular biology techniques has contributed to the identification of genomic mechanisms causing AMR. Recently, whole genome sequencing (WGS) enabled by next-generation sequencing (NGS) technologies has emerged as a powerful, rapid, and increasingly approachable tool for AMR prediction [5]. However, the application of WGS procedures can be challenging and is constantly evolving.

In the present study, we discus molecular mechanisms responsible for *H. pylori*’s AMR and the value of WGS approaches for tracking them. 

## 2. Antimicrobial Resistance in *H. pylori*

### 2.1. H. pylori Antimicrobial Resistance Worldwide

The increasing trend of AMR within *H. pylori* is of serious concern, particularly since it is the main reason for therapy failure. Although it is a worldwide emergence, there are geographical variations [4,6]. AMR is substantially higher in underdeveloped countries than in industrialized ones [7]. Along with other factors, the extensive use of antibiotics and the lack of *H. pylori* susceptibility testing in most laboratories are responsible for it. For instance, some authors maintain that high outpatient clarithromycin consumption in the treatment of respiratory tract diseases could account for its resistance increase [8,9]. Moreover, the observed higher prevalence of metronidazole resistance in women has been related to the prior use of metronidazole, particularly for gynaecological indications [10]. In addition, resistance to levofloxacin may be caused by the widespread use of this antibiotic for urinary and respiratory tract infections [11]. In contrast to the high and rising resistance rates in these antibiotics, in the case of resistance to amoxicillin, tetracycline and rifampicin rates remain quite low in most regions [12].

*H. pylori*’s AMR is mainly acquired by point mutations [13] (Table 1). Other possible mechanisms are loss of permeability through high expression of outer membrane proteins (OMPs) or loss of porins, overexpression of efflux pump transporters or biofilm formation [14]. In addition, *H. pylori*, in which clinical isolates exhibit increased mutation frequency, has a defective DNA mismatch repair system (lacking genes as *mutS*, *mutL,* and *mutH* implicated in this pathway). In addition, under stress conditions, the fact that it also lacks an SOS regulon induces recombination phenomena. Moreover, in the case of *H. pylori*, the active DNA polymerase I seems to be capable of fulfilling extra functions generating genomic plasticity. All this significantly contributes to the high genetic diversity of *H. pylori* isolates and it is associated with enhanced evolution of resistance to antibiotics [15,16,17,18].

It is worth pointing out that three resistance profiles can occur, (i) the one that affects a single antibiotic, (ii) multidrug resistance (simultaneous resistance to three or more antibiotics), and (iii) heteroresistance (a mixed population of susceptible and resistant organisms) [4,9,19,20].

Based on all the abovementioned, it is crucial to carry out further research on *H. pylori* AMR.

### 2.2. H. pylori Resistance Mechanisms

Clarithromycin

*Clarithromycin* is a macrolide antibiotic with bacteriostatic activity. Clarithromycin binds reversibly to the peptidyl transferase region of domain V of the 23S ribosomal subunit (23S rRNA), disrupting protein synthesis [13]. As opposed to other macrolides, clarithromycin can eliminate *H. pylori* due to its stability in acidic environments and its good absorption in the gastric mucus. 

The resistance of *H. pylori* to clarithromycin is mainly associated with point mutations in the V domain of *23S rRNA*. The most prevalent and well-recognized mutations occur at positions 2142 and 2143, including adenine to guanine transitions (A2142G and A2143G) and, less frequently, an adenine to cytosine transition (A2142C), in accordance with Taylor et al.’s numbering [13,21]. These mutations cause a change in ribosome structure resulting in inhibition of the binding between clarithromycin and the *23S rRNA*, interfering with protein synthesis and leading to a decrease in its efficiency [22,23]. 

Within the *H. pylori* genome, there are two copies of the *23S rRNA* operon. Some studies suggest that mutation in both copies of the *23S rRNA* gene is necessary for the strain to show resistance [13]. However, other authors have shown that a single copy might be sufficient to confer resistance [24].

Literature has linked more mutations to resistance to clarithromycin: A2115G, G2141A, A2144T, and T2289C. In addition, there are others that are not yet consistently reported or certain: G1939A, T1942C, G2111A, A2116G, C2147G, G2172T, T2182C, T2190C, C2195T, T2215C, A2223G, G2245T, C2245T, C2694A, and T2717C [8,22]. 

There are more genetic factors that could be involved in the increased AMR and have a synergic effect when coexisting with 23S point mutations. These factors include insertions or deletions in the *rpl22* gene (encoding a ribosomal protein that interacts with the 23SrRNA domains) and mutation at position 60 guanine to adenine in the *infB* gene (encoding a translation initiation factor, IF-2) [25]. 

Methylase synthesis and macrolide-inactivating enzyme activity have also been identified as resistance mechanisms. In fact, multidrug efflux pump transporters can also contribute to AMR in *H. pylori,* reducing intracellular antimicrobial concentration. Among efflux pumps of the resistance-nodulation-cell division (RND) family, the operon *hefABC* (hp0605–hp0607) has been linked to the development of clarithromycin resistance [26].

Metronidazole

Metronidazole is an antibiotic with high bactericidal activity that belongs to the family of nitroimidazoles. Metronidazole inhibits nucleic acid synthesis by its interaction with a nitroreductase homolog, rdxA. FrxA, another nitroreductase of *H. pylori*, may also activate metronidazole biocidal action. 

The mechanisms underlying resistance to metronidazole primarily involve the inactivation of nitroreductases responsible for producing antibacterial metabolites that damage bacterial DNA [27]. This inactivation is predominantly induced by complex genetic rearrangements in the *rdxA* gene, such as insertions and deletions, frameshift mutations, or missense and premature truncations [20]. 

Point mutation in the *frxA* gene has also been related to *H. pylori* metronidazole resistance, though the overall impact of *frxA* mutation is still uncertain. In addition, point mutations in the *frxA* gene as well as the *frxB* gene seem to be able to increase the level of metronidazole resistance in the presence of mutations in the *rdxA* gene [20]. 

Notwithstanding the above, the resistance to metronidazole is complex and still not well established since it may occur without the involvement of rdxA or frxA inactivation. As an example, amino-acid mutations (C78Y and P114S) of a mutant-type ferric uptake regulator (*Fur*) have been implicated due to an overexpression of the Sob enzyme, which is essential for the antioxidant defense of the bacteria. In addition, the drug efflux system from the RND family, such as HP0605 (*hefA*) and HP0971, may have a role to play in metronidazole resistance [28].

Levofloxacin

Levofloxacin is a third-generation fluoroquinolone. Levofloxacin acts as a bactericide by targeting chromosome replication, particularly with the inhibition of the DNA gyrase, since *H. pylori* lacks topoisomerase. 

Levofloxacin resistance in *H. pylori* has been associated with point mutations occurring in *gyrA* and *gyrB* genes that encode DNA gyrase subunits. Mutations in codons 87 and 91 of the quinolone-resistance determining region (QRDR) within *gyrA* gene (subunit A) are the most frequent in addition to being the best established [12]. At position 87, the change is asparagine to lysine, isoleucine, tyrosine, or histidine (N87K/I/Y/H), or threonine to tyrosine (T87Y). At position 91, the amino acid change is caused by the mutations is aspartate to glycine, asparagine, tyrosine, or alanine (D91G/N/Y/A). Mutations at positions 86, 88, 97, and 191 have also been described [27]. 

Less often, mutations of the *gyrB* subunit at positions 438, 481, and 484, or ejection pumps as well as loss of permeability, may be implicated [27].

Tetracycline

Tetracycline is a bacteriostatic agent that reversibly binds to the 30S subunit of *H. pylori* ribosomes containing *16S rRNA* and interferes with protein synthesis [29]. 

*H. pylori* resistance to tetracycline is rare and involves mutations in the primary binding site of tetracycline including single, double, and notably simultaneous triple base pair substitutions within both copies (*rrnA/B* genes) of *16S rRNA*, particularly at positions 926–928 [30]. 

The existence of *H. pylori* strains without mutations in the *16S rRNA* gene suggests a multifactorial resistance with other mechanisms being involved, such as reduced membrane permeability, alterations in ribosomal binding, enzymatic degradation of antibiotics, and an active efflux [22].

Amoxicillin

Amoxicillin is a β-lactam and bactericidal antibiotic. Amoxicillin inhibits bacterial wall synthesis by interfering with the synthesis of peptidoglycan. 

Resistance to amoxicillin in *H. pylori* is mainly attributed to mutations in the penicillin-binding proteins PBP1, PBP2, and PBP3 [31,32]. The PBPs are a group of proteins characterized by their affinity to b-lactams that take part in the synthesis and maintenance of the peptidoglycan layer of the bacterial cell wall. The decreased affinity to amoxicillin due to mutations in *PBP1A* has been shown to be the most implicated resistance mechanism. Unlike the others, PBP1 is a high-molecular-weight PBP with both transglycosylase and transpeptidase activity [32,33]. A specific feature of *H. pylori* is the absence or very rarely detectable β-lactamase activity. 

Some authors advocate that the determination of amoxicillin resistance should not be identified based on an amino acid at a specific position [24].

Other genes that could play a role contributing to resistance when mutations in the *PBP1A* gene are present are mutations in the *hopC* gene and the deletion in the porin-encoding *hopB* gene of *H. pylori* [34].

Rifampicin

Rifampicin and other drugs of the rifamycin group, rifabutin and rifaximin, are bactericidal antibiotics that inhibit transcription binding to the beta subunit of bacterial DNA-dependent RNA polymerase, which is encoded by the *rpoB* gene [12]. 

Rifampicin resistance is infrequent and mostly due to point mutations in the *rpoB* gene that causes lower affinity to the antibiotic. In *H. pylori*, resistance to rifampicin has been associated with amino acid exchanges in the rifampicin resistance-determining region (RRDR) of the *rpoB* gene, mainly at codons 525 to 545, 547, and 586 [35]. 

Rifampicin-resistant *H. pylori* strains have been described without any mutations in *rpoB*, suggesting that more mechanisms may be involved [22].

## 3. WGS

### 3.1. WGS Concepts

Because organisms, vegetable, bacteria, or mammal, have a unique genetic code, or genome, understanding the sequence of the bases in an organism, in other words, their sequence, can provide us with a great deal of information. Hence, DNA sequencing methods are laboratory procedures that determine the order of bases in the genome of an organism in one process. Compared with traditional DNA sequencing, NGS techniques have been a revolution that has allowed massively parallel sequencing of millions of DNA molecules. 

Given the parallel advancements in computation technology and software, WGS now allows us to determine nearly the entirety of the DNA sequence of an organism, both in humans and in microbes. In recent years, the time and cost of generating genome information has significantly declined [36]. Once the sequences have been obtained, researchers can either build a new genome from unknown organisms by using de novo sequencing with an assembly of the fragmented reads of DNA together, or measure genetic variation from an organism by comparing it to a reference genome. All of this enables us to see the presence or absence of certain genes, single-nucleotide polymorphisms (SNPs), structural differences, and other variations using several software programs [37]. 

Multiple technologies and platforms have been developed to carry out WGS studies [36]. In addition, a variety of tools are available to make these technologies accessible to researchers at laboratories around the world. For example, assembly tools such as Velvet or SPADES; genome characterization tools such as KmerFinder, NCBI BLAST, PROKKA, or PlasmidFinder among others; or comparative genomic tools such as PubMLST, NDtree [36,38]. To facilitate use by those who lack sufficient bioinformatics knowledge, easy-use software tools have been designed. 

### 3.2. WGS in Clinical Microbiology

Nowadays, microbiologists, clinicians, and researchers can obtain the complete sequence of microorganisms, both chromosomes as well as all extrachromosomal mobile genetic elements (MGEs), plasmid DNA or bacteriophages, if any. 

Traditionally, the main use of WGS techniques has been the study of bacteria. Nevertheless, in recent years and, in a remarkable way, after the appearance of the SARS-CoV-2 virus, these tools have also undergone a great expansion in the virology field. 

Applications of WGS to the genome of bacteria are valuable for identification, typing, resistance detection, study of virulence factors, surveillance, and epidemiology [39]. In addition, they have also supported AMR prediction, evolutionary analysis of infectious diseases, and investigation of food-borne disease outbreaks related to the origin of pathogen strains [38].

### 3.3. WGS for H. pylori Resistance Detection

Genotypic characterization of isolates is crucial to further investigate the mechanism underlying AMR. Since the emergence of *H. pylori* resistance has been mainly related to mutational changes within the antimicrobial target genes, WGS-based methods might be of major benefit [40]. 

Phenotypic and genotypic correlation has proved to be accurate for clarithromycin (*23SrRNA* gene), levofloxacin (*gyrA/B* genes), and amoxicillin (*PBP1A* gene). In the case of metronidazole, its complex mechanism of resistance makes further research necessary [41]. Finally, both rifampicin and tetracycline have such low resistance rates that it has not yet been possible to establish definite correlations [22].

Not only spontaneous point mutations associated with AMR and AMR genes can be identified through WGS data, but also novel or rare resistance mechanisms [25,38,42]. This would enable us to reanalyze retrospectively according to new discoveries.

On one hand, other possible *H. pylori* mechanisms can be studied with WGS procedures, including MGEs, high expression of OMPs, efflux pump transporters, or sporadic mutations. On the other hand, relationships between the presence or absence of certain virulence factors and resistance mechanisms can be studied in order to establish some possible correlations [14,29].

With WGS, multiple sequencing reads are obtained per sample. Therefore, it is also applicable to the detection of heteroresistance within *H. pylori* [12]. Moreover, owing to the presence of two copies in the *23SrRNA* gene, WGS can be a powerful technique to examine the depth of reads mapped to each nucleotide variation position known to be associated with resistance [43].

Plasmids are not commonly found when investigating *H. pylori* [5,11], and neither are horizontally acquired MEGs with impact on AMR, inspiring us to think that there is no association between MEGs and resistance within *H. pylori* [29].

*H. pylori* presents an enormous genetic diversity because of high mutation and recombination rates, which at times complicates its interpretation [24].

### 3.4. WGS Bioinformatics Analysis and Online Platforms

A wide variety of methods and tools are available to analyze and characterize bacterial pathogens directly from the data produced by widely used sequencing technologies. Many of them require some bioinformatics expertise to be implemented for high-throughput environments, such as UNIX. 

Limited resources for the bioinformatic analyses could become a major obstacle. For this reason, online platforms as well as commercial software tools are available, commonly free of charge, for those who lack or have limited bioinformatics knowledge. These web-based tools are equipped with their own set of pipelines and need to be kept up to date for their use in the real-world scenario [38].

Nevertheless, not many of the bioinformatic tools and databases can be applied to WGS data aiming to identify the AMR genes of interest, point mutations, or associated MGEs.

When working with sequencing data to detect resistance mechanisms, most of the available tools fall into one of two categories: (a) assembly-based tools that align sequencing reads to a reference genome, usually *H. pylori* reference strain 26,695 (NCBI reference sequence: NC_000915.1), and (b) read-based tools that match sequence reads to a reference gene to identify gene presence (de novo assembled sequences) [44].

(a) In the assembly-based approach, sequences of previously assembled and annotated genes can be compared to a set of reference genes using BLASTN-based tools. While other bacterial species have well-established comprehensive AMR data, such us AMR Gene-ANNOTation (ARG-ANNOT), MEGAres, Resistance Gene Identifier (RGI) searching the Comprehensive Antimicrobial Resistance Database (CARD), or ResFinder, these databases do not include much, if any, *H. pylori* information [24]. CGE ResFinder v.4.1 (now including PointFinder) is a web and standalone tool and database that identifies acquired genes and/or finds chromosomal mutations. However, it is able to detect single nucleotide polymorphism (SNP)-related resistance, so it requires a specific database for *H. pylori* mutation detection, which is still under development [45]. RGI v.6.0.1, which can detect SNP-related resistance, is a web and/or standalone tool searching CARD v.3.2.6 that is useful in the identification of point mutations among the most well-known resistance genes within *H. pylori*. It does not use a specific database for *H. pylori*, so it is necessary to previously know about the implication of the mutations found. CRHP Finder is a webtool for clarithromycin-resistance prediction based on the detection of *23S rRNA* point mutations (A2142C, A2142G, A2143G, T2182C, C2244T, T2712C). CRHP Finder has been recently validated but is not yet available [43].

(b) In read-based approaches, short reads are either aligned to reference databases through pairwise alignment tools or break into shorter k-mers subsequently mapped to a k-mer database obtained from reference sequences. Some examples include KmerResistance, PATRIC, or older versions of ResFinder as v.3.2 [38]. These methods are less computationally demanding, as assemblies are not required. While these approaches can detect resistance genes, they do not address chromosomal mutations associated with AMR. Using this approach, the BV-BRC system (combined with PATRIC), based on de novo-assembly, integrates bacterial information via thousands of bacterial genomes with analysis. In addition, it provides open-source tools for *H. pylori* data analysis and genomic annotation. It is useful in the detection of resistance genes, protein, structure and function, and other features including virulence factors or efflux systems. However, it fails to detect point mutations, which means an obstacle for *H. pylori* study [46]. At the time of preparation of this article (May 2023), a total of 2318 whole genome sequences of *H. pylori* isolates were available from all over the world in the BV-BRC 3.30.5 database.

### 3.5. Advantages

NGS techniques are highly sensitive for detecting the presence of mutations and resistance genes. When compared to other techniques, the study of *H. pylori*’s AMR with WGS procedures can be of benefit. On the one hand, WGS techniques have already been applied directly from clinical specimens, providing an attractive option [41,47], since culture of *H. pylori* is not always successful due to the growth requirements of the bacterium. For this purpose, excellent DNA extraction methods are required, and progress is being made in this field. On the other hand, because PCR techniques are limited to specific targets, this can lead to false negatives. 

AMR study using WGS is not only useful to assist clinicians in the treatment of a specific patient, but also to promote the progress of research on the molecular mechanisms of resistance.

In addition, along with the development of new approaches and bioinformatics tools to analyze and extract the relevant genomic data, a rich area of work has been generated thanks to publicly available genomes on online platforms and databases. Therefore, WGS data can be used to compare genomes from different regions of the globe. Another advantage is the potential of WGS for retrospective studies when new mechanisms involved in resistance are discovered.

### 3.6. Pitfalls

Current limitations of WGS for *H. pylori* AMR study are:-Technical limitations of NGS techniques such as coverage, depth, and read length. Also, quality assessment of draft genomes is crucial in order to avoid contamination and heterogeneity of the genomes.-As different pipelines are used to generate assemblies, certain heterogeneity of the resulting contigs may appear.-There is a need for a deep understanding of how closely genetic determinants correlate with phenotypic resistance, to give accurate value to the sequence information obtained. This is still a disadvantage regarding metronidazole, tetracycline, and rifampicin.-Inability to determine minimum inhibitory concentrations (MIC) through NGS techniques.-Possible underestimation of resistance mechanisms not represented in databases because of (a) nongenetic mechanisms involved or (b) emergence of novel mutations, since *H. pylori* AMR is mainly produced by non-inheritable (mutational acquired) resistance.-Imperative need for continuous updating of the databases and their users.

## 4. Discussion

The emergence of AMR leads to treatment failure, particularly in some regions.

AMR to *H. pylori* is primarily due to chromosomally genetic changes, with no extrachromosomal elements or plasmid involvement. In addition, vertically transmitted point mutations in the DNA are the foremost cause, as opposed to resistance gene acquisition. Efflux pumps are also involved in AMR within *H. pylori*, though their role is not as well established, and the mechanism of acquisition is still unclear. 

Genomic insight into AMR mechanisms and appropriate molecular tools are keys to optimizing treatment strategies and updating guidelines according to the epidemiology. The fact that chromosomal mutations are mainly responsible for resistance makes it much easier. Nowadays, a good correlation between the genetic findings and the response to treatment has been established for some antibiotics, especially for clarithromycin, levofloxacin, and amoxicillin. Yet still, no determinants are undoubtedly genetically linked to metronidazole, tetracycline, or rifampicin.

Since NGS methods can quickly and accurately provide sequence information, WGS has become an available method that allows us to carry out more in-depth studies and better understand the mechanisms of resistance. More platforms and databases are desirable so that we can accurately interpret all the information from the *H. pylori* genomes to provide significant decision-making information for microbiologists, clinicians, and public health policy makers.

## 5. Future Directions and Personal Assessment

In the context of difficulties with bacterial culture and since antimicrobial susceptibility testing is not routinely performed in the clinical practice, there has been an increase in treatments without sensitivity testing. Alternate studies on the resistance status of *H. pylori* are needed. Hence, given the democratization of molecular biology systems in medical laboratories, applications for the study of AMR in *H. pylori* should be mandatory.

The potential of WGS has already been assessed to identify genetic AMR-related determinants of *H. pylori* associated with therapeutic failure, even though monitoring of the already-known mechanisms and further insights are necessary to better understand and establish the mechanisms of resistance for tetracycline, rifampicin, and metronidazole. In addition, in order to combat AMR, WGS techniques could also be of value for the identification of *H. pylori* membrane protein receptors, which might contribute to the design of therapeutic drugs and vaccine development [48]. 

Nonetheless there is a need for implementing WGS approaches and pipelines for *H. pylori*. In addition, bioinformatics tools ought to be simpler. Particularly, these tools should be improved for *H. pylori*, taking other bacteria as *Campylobacter spp*. as an example. Standardization of procedures as well as validation of databases to ensure the comparability of WGS data and results between laboratories are required. 

On the other hand, as diagnoses of *H. pylori* by non-invasive techniques are on the rise, monitoring of AMR studies through WGS techniques should be handled periodically and regionally to define the evolution of the resistance patterns of *H. pylori* and provide direction for guidelines or protocols. Choosing a regimen based on local susceptibility patterns or, even, on the susceptibility profile of an individual, would prevent unnecessary antimicrobial exposure and optimization of treatments. 

All this leads to the assertion that effective eradication programs of *H. pylori* in the near future will be based on WGS approaches.

## Figures and Tables

**Table 1 antibiotics-12-01135-t001:** Summary of the main mechanisms of antibiotic resistance and targeted genes and mutations.

Antibiotic (Drug Class)	Mechanism of Action	Main Resistance Mechanism	Specific Mutations
Clarithromycin (macrolide)	Inhibition of the protein synthesis through interaction with the ribosome machinery (V domain of *23S rRNA*)	Point mutations in the V domain of *23S rRNA*	A2142G, A2143G, A2142C
Metronidazole (nitroimidazole)	Inhibition of the nucleic acid synthesis by interaction with a nitroreductase homolog, *rdxA*	Insertions/deletions, frameshift mutations or missense and premature truncations in the *rdxA* gene	
Levofloxacin (fluoroquinolone)	Inhibition of the DNA gyrase by interfering with DNA	Point mutations in the QRDR within *gyrA* gene	Positions 87 and 91
Tetracycline (tetracycline)	Inhibition of the protein synthesis through interaction with the 30S subunit of ribosomes (*16S rRNA*)	Triple base pair substitutions of *16S rRNA*	Positions 926–928
Amoxicillin (β-lactam)	Inhibition of bacterial wall synthesis by interfering with the synthesis of peptidoglycan	Mutations in the *PBP1A*	
Rifampicin (rifamycin)	Inhibition of transcription binding to the beta subunit of bacterial DNA-dependent RNA polymerase encoded by the *rpoB* gene	Point mutations in the RRDR of the *rpoB* gene	codons 525 to 545, 547, and 586

## Data Availability

Not applicable.

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
