# Peer review of "Whole Genome Sequencing for Studying Helicobacter pylori Antimicrobial Resistance"

_antibiotics, 2023, doi:10.3390/antibiotics12071135_

Round 1

Reviewer 1 Report

Authors have written a perspective on the role of WGS in enabling the detection of specific genetic mutations or acquired resistance genes that confer antimicrobial resistance in H. pylori. These mutations or genes can be linked to resistance against antibiotics commonly used in H. pylori treatment, such as clarithromycin, metronidazole, and levofloxacin.

The perspective is well-written and the organization is clear.

I have a suggestion though for the following chapter:

2.2. H. pylori resistance mechanisms 

I would like to see a summary table including antibiotic, Drug Class, target, mechanism,

specific mutations (check CARD updated), structural consequence (if known),

any other relevant info.

English is fine.

Reviewer 2 Report

In this review, the authors summarized the resistance mechanisms of H. pylori and the application of WGS in the AMR of H. pylori. However, I have several issues should be addressed.

1. In this article, the authors mentioned that WGS has the potential to overcome the challenge of H. pylori culture. However, in my understanding, WGS also needs to culture H. pylori and then extracts DNA from the culture for sequencing.

2. The title “WGS bioinformatics analysis vs online platforms” of section 3.4 was not appropriate, as the content mainly introduced the available online platforms, rather than comparing WGS bioinformatics analysis and online platforms.

3. The article should summary the current research progress in studying H. pylori AMR with WGS.

The quality of English language is acceptable.

Reviewer 3 Report

We are rapidly progressing towards the post antibiotic era and therefore a thorough understanding of various mechanisms of AMR in bacteria is one of the pressing issues in current research scenarios.

H. pylori has been classified as priority for investment in new drugs by WHO. In this review, authors wish to discuss the use of whole genome sequencing (WGS) to study antibiotic resistance in H. pylori.  Although the topic of this review is very important, the manuscript do not provide sufficient examples of detection of AMR using WGS, appropriate references are not provided, some sections are repetitive and simplistic (in my opinion).

Major and minor points:

Authors could discuss in more detail about the prevalence of antibiotic resistance in H. pylori.

Various molecular mechanisms of AMR in H. pylori such as high mutations rates, together with presence of error prone DNA polymerases, lack of mismatch repair system, high rates of natural transformation should be discussed with appropriate acknowledgements.

Line 69: the sentence seems incomplete.

I wonder what authors are trying to discuss in lines 177-217. They have discussed the complex mechanisms of antibiotics resistance in the first part of the manuscript. Such a simplistic explanation seems repetitive and unnecessary at this point. Moreover, appropriate references have not been provided.  This section seems to be a collection of random repetitive sentences.

Authors should provide specific examples of how AMR were identified using WGS.

They should also mention what novel AMR mechanism have been identified using WGS rather than directing to the reference (line 230)

Line 249 : please name at least few methods and tools

Line 299-300 : most of the H. pylori strains are cultivable. If not, appropriate reference should be provided.

Lines 261-294 are informative; I think the author should provide more details in this section.

Section 3.6 One of the pitfalls could be non-inheritable antibiotic resistance.

The authors could discuss the implications of WGS for the development of new drugs and vaccines to combat AMR.

The English could be improved. Although I don’t consider myself an expert in the English language, there are several language errors and incomplete sentences in this manuscript. Please proofread the language before submission.

English need moderate editing

Round 2

Reviewer 3 Report

Thanks for incorporating my suugesttions. This topic is highly relevant and therfore I would like to thank authors for their efforts.

Author Response

Thanks for helìng us to improve the quality of the manuscript